# On the Tensile Behaviour of Bio-Sourced 3D-Printed Structures from a Microstructural Perspective

**DOI:** 10.3390/polym12051060

**Published:** 2020-05-06

**Authors:** Sofiane Guessasma, Sofiane Belhabib, Abdullah Altin

**Affiliations:** 1INRAE, UR1268 Biopolymères Interactions Assemblages, F-44300 Nantes, France; 2IUMR CNRS GEPEA, Université de Nantes, Oniris, CNRS, GEPEA, UMR 6144 F-44000 Nantes, France; sofiane.belhabib@univ-nantes.fr; 3Department of mechanical and metal technology, University of Yuzuncu Yil, Van Vocational of Higher School, 65100 Van, Turkey; aaltin@yyu.edu.tr

**Keywords:** fused deposition modelling, PLA tensile properties, X-ray micro-tomography, finite element computation, infra-red measurements, high-speed camera, damage modelling, printing temperature, thermal cycling, microstructure

## Abstract

The influence of the microstructural arrangement of 3D-printed polylactic acid (PLA) on its mechanical properties is studied using both numerical and experimental approaches. Thermal cycling during the laying down of PLA filament is investigated through infra-red measurements for different printing conditions. The microstructure induced by 3D printing is determined using X-ray micro-tomography. The mechanical properties are measured under tensile testing conditions. Finite element computation is considered to predict the mechanical performance of 3D-printed PLA by converting the acquired 3D images into structural meshes. The results confirm the leading role of the printing temperature on thermal cycling during the laying down process. In addition, the weak influence of the printing temperature on the stiffness of 3D-printed PLA is explained by the relatively small change in porosity content. However, the influence of the printing temperature on the ultimate properties is found to be substantial. This major influence is explained from finite element predictions as an effect of pore connectivity which is found to be the control factor for tensile strength.

## 1. Introduction

Polylactic acid (PLA) is by far the most studied biosourced feedstock material for polymer-based additive manufacturing [1,2,3]. PLA is derived from starch by mechanisms of hydrolytic cracking and fermentation. The use of PLA in additive manufacturing processes such as Fused Deposition Modelling (FDM) has many advantages. PLA exhibits a remarkable ability for thermo-forming under low processing temperatures compared to other feedstock polymers. It has also rapid crystallisation upon cooling. Besides the processability aspects, PLA is biodegradable, renewable resource, affordable and exhibits a low toxicity compared to ABS (acrylonitrile butadiene styrene) that is also widely used in FDM. The printing conditions of PLA have been extensively studied and these were related to both the thermal and mechanical behaviour of the raw polymer. The thermal kinetics during printing especially the consequent crystallisation behaviour of PLA is significantly affected by its tacticity and chemical composition. In this regard, the influence of the glass transition, the degree of crystallinity, and melting temperature were investigated. It is well known that printing parameters that influence the thermal cycling during the printing process have a significant effect on the polymer relaxation trend. This is particularly true when the semi-crystalline nature of feedstock PLA leads to slower crystallization behavior compared to the rapid thermal cycling undertaken during the laying down process [4]. In this regard, Lee et al. [5] investigated the quality of 3D-printed PLA for a varied air-cooling rate. The authors showed that this parameter influenced significantly both the mechanical strength and the dimension accuracy. Liao et al. [6] investigated the crystallinity and porosity of 3D-printed PLA using X-ray diffraction. The authors showed that differences were observed between thin and thick samples because of different thermal histories. While thin samples did not come up with significant variations, the thick ones revealed different crystallization trends depending on the base temperature. In particular, it was found that the presence of δ crystallite form (less stable than the α form) decreased the stiffness of printed PLA. In addition, Pandis et al. [7] reported that the increase in crystallinity is found after several days of in-service for 3D-printed PLA. The authors attributed this phase change to material aging. In terms of mechanical performance, tensile, compression, flexural, impact and fatigue behaviour of PLA were correlated to varieties of printing parameters such as the infill density, infill pattern, and printing orientation to name a few [2,8]. One particular aspect that received major attention is the interfacial interactions during the printing process [9], which prove to be correlated to scale dependence of the mechanical properties of the printed parts. In a former study [10], the authors showed that the scaling up of 3D-printed polymeric structures (typically from 5 mm up to 40 mm) results in a decrease of the mechanical performance. The authors attributed this performance decrease to larger filament paths that increased the thermal cycling duration. As a consequence, the rapid cooling would not allow a more cohesive structure to form because less stable crystalline forms emerge. A lower cooling rate would be more beneficial for the mechanical integrity of the 3D-printed part. In addition, X-ray micro-tomography results clearly indicate that the porosity arrangement within the printed structure exhibits a significant anisotropy depending on the thermal history [11]. The porosity profiles along the printing directions differ depending on the raster and the filament laying down conditions (nozzle speed, printing temperature). Another aspect related to the significance of the interfacial interactions is the dependence of the mechanical properties on the layer thickness. Several authors addressed this issue, which led to the well-known recommendation of using a layer thickness as a fraction of the nozzle diameter (half of the nozzle diameter is generally used). For instance, Chacon et al. [12] studied the effect of layer thickness on the mechanical performance of 3D-printed PLA. The authors concluded on a contrasted effect of this parameter with regards to the part orientation. The most significant effect was related to the ductility decrease when the layer thickness was increased. Because of the importance of the interfacial interactions, several studies demonstrate that post treatment of 3D-printed parts improves the mechanical performance. For instance, Shaffer et al. [13] reported that improved interlayer adhesion is obtained when the 3D-printed part is exposed to ionizing radiation. This treatment contributes to enhance the polymer crosslinking. Furthermore, Bhandari et al. [9] showed that annealing treatment with temperatures higher than the glass transition temperature improves the inter-layer tensile strength. 

Because of some issues related to the use of pure PLA in 3D printing, such as lack of thermal stability, brittleness and degradation behaviour, several attempts were made to blend the filament with different materials such as lignin [14], hemp fibres [15], poly ε-caprolactone [16], wood flour [17], and carbon whiskers, among other fillers. 

The abundant literature work on PLA has also covered topics related to numerical modelling including Finite Element (FE) models. The numerical developments focused on how to implement raster information in mechanical models to predict the mechanical performance of 3D-printed parts [18]. In this study, both experimental and numerical approaches are considered to reveal the influence of the microstructural arrangement of PLA filaments on the tensile behaviour of printed parts. In particular, the 3D imaging technique is used to quantify the microstructural defects in printed samples. The same images are also used to feed a mechanical model that explicitly implements the microstructural information and predicts the deformation mechanisms related to the filament arrangement. The approach considered in this study that relies on exploring the microstructural effects on the mechanical performance is a rather new trend in additive manufacturing [19]. Early contributions focused mostly on building experimental designs to establish correlations between the process parameters and the mechanical performance [20]. These approaches have several outcomes, such as the quantification of the parameter interdependence, recommendations for successful prints, and fast prediction of part performance. However, these, at the same time, have some oversights because the reading of the physical explanations from these correlations is not clearly obvious. 

## 2. Experimental Layout

The PLA wire is used as a feedstock material for FDM experiments. The PLA wire of 1.75 mm in diameter is purchased from Sprinteo company (Eaunes, France). The supplier recommends an operating temperature of 190 °C. Differential Scanning Calorimetry (DSC, Mettler Toledo, Columbus, OH, USA) experiments are used to assess the glass transition during heating and cooling stages. These experiments are conducted using DSC823e equipment from Mettler Toledo (Viroflay, France) The glass transition is screened within a temperature range between −40 and 210 °C. The sample weight is 13 mg and the heating and cooling rates are 10 °C/min. The first scan is used to assess the glass transition of the as-received PLA filament. The second scan anticipates the thermal changes after erasing the thermal history of the filament. 

PLA samples are printed using Replicator 2 from MakerBot (Figure 1a) that uses fused deposition modelling (FDM) technology. The process conditions are: 0.4 mm for nozzle diameter, +45°/−45° for layups, building direction // sample thickness, 0.2 mm for layer height, 100% for infill, 150 mm/s for nozzle vertical speed, raft option activated, printing temperature varied between 210 and 255 °C. Although, the recommended printing temperatures for PLA are below 230 °C, the objective of using larger temperatures is to assess the effect of large temperature ranges on mechanical performance that are not yet reported. This allows to establish a more precise parameter window for PLA printing.

Main dimensions of printed dog-bone specimen are shown in Figure 1a. Computer Aided Design (CAD) models are prepared using MakerBot© Desktop software (version 3.0, MakerBot Industries, NY, USA). Both surface tessellation and conversion to toolpaths are performed with this software. 

Weight and dimensions of printed samples are measured to assess volume mismatch and overall density. The FDM process is also monitored using infrared (IR) camera from Flir company (Flir A35 series, Wilsonville, OR, USA). Sample geometry and setup are illustrated in Figure 2. The IR monitoring is conducted with a resolution of 320 × 256 pixels and a frame rate of 60 Hz. The calibration of IR results is conducted by a contact thermocouple and shows a discrepancy of less than 3 °C for temperatures as large as 190 °C. IR images are processed using Flir tool software (version Plus, Flir company, Wilsonville, OR, USA).

With four samples per condition and 30 minutes per sample printing, the printing campaign takes roughly one day. 

The 3D imaging of PLA samples is conducted using X-ray micro-tomography technique (UltraTom X-ray micro-CT from Rx-Solutions company, Chavanodcity, France). The acquisition parameters are as follows: 230KV X-ray source, 80KV for voltage, 480 µA for current intensity, 1440 for the number of radiographic images, 1920 × 1536 pixels for detector resolution, 14.31 µm for voxel size. The set of radiographic images is converted into a tomogram using a back-projection reconstruction algorithm using X-Act software from Rx-Solutions. The typical tomogram resolution is 1277 × 1043 × 438 voxels and an acquired volume of 18 × 15 × 6 mm^3^. Image analysis is conducted using ImageJ software from NIH (version 1.43, NIH, Bethesda, MD, USA). 

Tensile testing is conducted on both neat and notched 3D-printed PLA samples (notch depth of about 1 mm). In addition, as-received PLA wires are also submitted to tension tests. Mechanical characterisation is performed using Zwick Roell machine (Zwick Roell Group, Ulm, Germany) with a load cell of 10 kN, and a fixed displacement rate V=5 mm/min. Tensile testing is monitored using high-speed camera (Phantom V7.3 from Photonline company, Marly Le Roi, 78-France). Sample failure is observed at high magnification (128 × 80 pixels) and a large frame rate (76,000 to 111,000 fps, frames per second) while the entire tensile sequence is recorded using full resolution (800 × 600 pixels) and a small frame rate (100 fps). 

Filament arrangement and rupture patterns are analysed using a Scanning Electron Microscope SEM (JEOL JSM 7600F, Jeol Ltd, Akishima, Japan). Images are obtained with a resolution of 1280 × 1024 pixels and a pixel size between 0.11 and 3.13 µm. Sample preparation requires gold/palladium coating prior observation. 

## 3. Modelling Technique

3D images of 3D-printed PLA are implemented in a finite element model by converting the image voxels into 3D structural elements. The solid phase is regularly meshed according to a voxel-element conversion. Structural cubic shape elements are defined by eight nodes and three displacement components (UX, UY and UZ) per node. Both the sample size and the resolutions are varied to assess the effect of the microstructural details and representative elementary volume. Cubic volumes are collected from the central part of the imaged sample for an increased size from 0.4 mm up to 3 mm. The model size ranges from 0.77 × 10^6^ to 156 × 10^6^ dofs (degrees of freedom) depending on the resolution (voxel size between 14.31 and 143 µm) and sample volume (between 0.05 and 1136 mm^3^). 

An isotropic elastic material model is considered to assess the tensile behaviour of the 3D-printed PLA. Young’s modulus $E_{Y0}$ and Poisson’s coefficient $v_{0}$ of PLA are implemented according to experimental data and technical sheet provided by the supplier. 

Two modelling schemes are implemented. The first one is a quasi-static linear model that allows the determination of the stiffness of 3D-printed PLA based on microstructural information. For this model, tension is simulated in X-direction by blocking all lateral displacement components (UY = 0 and UZ = 0) on both ends (x = 0, x = L, where L is the sample length), imposing a given displacement amount UX = U > 0 at all nodes located at x = L, and fixing the same component (UX = 0) for nodes belonging to the other end (x = 0). 

The elasticity problem is solved using preconditioned Conjugate Gradient (PCG) solver and the overall Young’s modulus is predicted from the reaction force in X-direction at the loaded end (x = L). In addition, stress fields are derived to determine the role of the microstructure in stress heterogeneity. 

The second model is an elastic-damage model that has the objective to describe the damage evolution during the tensile loading. This damage model assumes a linear damage law with an onset determined by a critical engineering strain level $\varepsilon_{r}$ and a damage rate $D_{r}$. According to this model, Young’s modulus is decreased locally to a ground value $E_{G}$ to account for the consequent creation of material discontinuity or cracking at regions of high stress. This can be formulated as follows
(1)$$E_{Y0}\left(GPa\right)\overset{}{\to }E_{G}\left(GPa\right)=E_{D}\left(\%\right)\times E_{Y0}\left(GPa\right)\;|\varepsilon \left(\%\right)>\varepsilon_{r}\left(\%\right)\times Max\left(\sigma_{I}\right)$$
(2)$$d\left(\varepsilon \right)=D_{r}\left(\%\right)\times V\times \left(\varepsilon /\varepsilon_{R}\right)$$
where ε is the engineering strain, $\sigma_{I}$ is the stress intensity, $\varepsilon_{R}$ is the elongation at break, V is the sample volume, $E_{D}\left(\%\right)$ is the modulus reduction ratio at damaged zone. 

Due to the large models used, high performance computation resources are used with CPUs operated at 4.4 Ghz and 1 Tbytes of RAM. The computation duration reaches up to 280 min per load increment. 

## 4. Results and Discussion

### 4.1. Thermal Behaviour

Figure 3 shows that DSC measurements suggest, at the first heating stage, a glass transition *T*_g_ of PLA of 62 °C and melting temperature around 157 °C. Kim et al. [21] reported a similar *T*_g_ value (62.9 °C) and a slightly larger melting temperature (167.2 °C) for PLA similar to the one considered in this study. The presence of single peaks close to the relaxation and melting points indicates a clear *T*_g_ and simple melting event. During the cooling stage, *T*_g_ is captured around 51 °C and there is no evidence of crystallisation. At the second heating run, *T*_g_ is close to 58 °C.

Figure 4 shows infra-red signature of the printing nozzle during the filament laying down captured at two distinct moments. Despite the limited resolution of the camera and the zoom-out views, the snapshots suggest a rapid cooling of the filament temperature at the nozzle exit. This can be read from the large contrast between the nozzle temperature (290 °C) and the temperature of the laid down filament (below 100 °C). However, the snapshots do not show clearly the temperature level variations between printing conditions. For such comparison, quantitative data extracted from the temperature measurements are used. 

The laying down process is continuous as there is no sudden change in the temperature along the printing path. Figure 5 shows more quantitative data about heat accumulation during the printing process. To build the thermal cycle profiles in Figure 5a, the temperature reading of the IR camera at a particular position is monitored as a function of time. The point selected for this reading lies at the middle position. At each nozzle pass, the temperature increases to its peak value when the nozzle position is coincident with the point position. When the nozzle moves away, the temperature at the sensed position decreases. The succession of heating and cooling stages at the selected position builds up the thermal cycling curves depicted in Figure 5a. These thermal cycles are informative about the kinetics of heating and cooling during the filament laying down. 

In particular, Figure 5a shows the thermal cycling for two extreme printing temperatures. Although, the low limit for printing temperature used in this study (210 °C) is commonly used for PLA, high processing temperatures such as 255 °C may result in PLA degradation process. Coppola et al. [22] conducted a thermogravimetric analysis (TGA) up to 700 °C, which shows weight loss values of 25% at temperatures starting from 346 °C. Their results suggest that the thermal stability of PLA is not compromised at 255 °C. Silverajah et al. [23], in an earlier work, report a slightly lower degradation temperature for PLA of 325 °C. 

Large periods between the first peaks indicate the beginning of the raft stage where the structure is less compact and is associated with the junction between the building platform and the part. The distance between peaks become narrower starting from 120 s and resulting from the building of the part. Only slight changes are observed between ground temperatures for both high and low printing temperatures. The overall ground temperature is close to 36 °C. The linear correlation between the printing temperature $T_{P}$ and the ground temperature $T_{G}$ is as follows:(3)$$T_{G}\left({}^{\circ}C\right)=23+0.06\times T_{P}\left({}^{\circ}C\right);\;R^{2}=0.83$$

Most of the variation is related to the peak values, where it is found that the increase in the printing temperatures is correlated to larger peak temperatures $T_{K}$. This correlation is expressed in form of a linear function
(4)$$T_{K}\left({}^{\circ}C\right)=-35+0.74\times T_{P}\left({}^{\circ}C\right);\;R^{2}=0.83$$

In addition, it is found that the average cooling rate increases from 5.85 to 7.59 °C/s when the printing temperature is increased from 210 to 255 °C. 

### 4.2. Mechanical Behaviour

The tensile response of as-received PLA filaments is depicted in Figure 6. Figure 6a shows a large stretching level prior rupture. The stress–strain curves for three replicates are shown in Figure 6b. 

The overall filament behaviour is an elastic stage followed by a large plastic plateau with elongation at break as large as 70%. Table 1 summarised the main tensile quantities for the PLA filament. The observed tensile strength of PLA is within the range reported in the literature for PLA (48–60 MPa) [24,25]. 

The measured Young’s modulus $E_{Y0}$ is lower compared to what is usually reported for PLA, but this is more related to the testing conditions, in particular the small displacement rate used in this study. The dispersion of the results is acceptable for tensile strength $\sigma_{S0}$ (0.3%) and stiffness $E_{Y0}$ (13%), and larger for the elongation at break $\varepsilon_{R0}$ (33%). 

Figure 7 illustrates the recorded deformation sequences by a high-speed camera of 3D-printed PLA under various conditions. For neat specimens (Figure 7a), the increase in the printing temperature from 210 to 230 °C does not lead to a significant change in the rupture scenario. 

The rupture seems to be less correlated to the raster or filament orientation and the printed PLA seems to follow a dominant opening mode with a limited stretching at the rupture point and no evidence of section reduction. Zoom-in views of notched specimens under tension (Figure 7b) confirm the dominance of the opening mode, where crack propagation monitored at high recording speed demonstrates the minor change in the crack path up to the rupture point. At ultra-high speed, with frame times as small as 9 µs, the recorded crack speed provides a low estimation of 440 m/s (Figure 7c). 

Figure 8 shows the tensile response of both neat and notched specimens for three printing temperatures. These curves confirm the small elongation at break of the printed PLA compared to the as-received filament. 

There is also a slight improvement of both tensile stiffness $E_{Y}$ and strength $\sigma_{S}$ with the increase in the printing temperature. This improvement is more or less balanced by the decrease in the elongation at break. The correlation between the tensile quantities and the printing temperature can be approached using linear fitting functions according to the data reported in Table 1.
(5)$$E_{Y}\left(MPa\right)=252+2.35\times T_{P}\left({}^{\circ}C\right);\;R^{2}=0.93$$
(6)$$\sigma_{S}\left(MPa\right)=14+0.17\times T_{P}\left({}^{\circ}C\right);\;R^{2}=0.96$$
(7)$$\varepsilon_{R}\left(-\right)=17-0.04\times T_{P}\left({}^{\circ}C\right);\;R^{2}=1.00$$
(8)$$K_{1C}\left(MPa\times m^{1/2}\right)=-4.73+0.03\times T_{P}\left({}^{\circ}C\right);\;R^{2}=0.62$$
where $K_{1C}$ is the fracture toughness.

The discrepancies of the results in Table 1 are acceptable and confirm the reproducibility of the printing condition. The average scatter for Young’s modulus, tensile strength, and elongation at break is 4.7%, 2.7%, and 5.3% respectively. The exception is the fracture toughness where the scatter is found close to 24%. 

The comparison between the as-received and printed PLA performance shows that there is a loss of mechanical performance that can be also correlated to the printing temperature. Figure 9 provides a qualitative explanation for this difference based on the load transfer schematic for a typical 3D-printed structure. The first difference is in the deformation modes. For a raster with a pill ups of −45°/+45°, the deformation of the printed structure combines both shearing and tension while the as-received material is only subject to pure tension. In addition, the voids left within the raster act as stress concentrations and result in lower stiffness and strength. The presence of such voids is responsible for the stress heterogeneity in the printed PLA. In contrast to this stress heterogeneity that develops in the entire 3D-printed part, the stress distribution within the as-received wire subject to tensile loading is more homogeneous if the stress concentration at the grips is excluded. Finally, the particular arrangement of the filaments in the 3D-printed part allows only a limited part of the filament to be stretched, which also causes inhomogeneous deformation. At these locations, the PLA filament deforms up to its elongation at break. This effect is more amplified for filament with larger ultimate properties such as copolyester [26]. By opposition to this varied streching, the tensile loading of the as-received wire is uniform along a large part of its length. 

Based on the correlations described by Equations (3) to (5) and the data reported for the as-received PLA, the following tendencies for mechanical loss are identified.
(9)$$\Delta E_{Y}/E_{Y0}\left(\%\right)=-77+0.22\times T_{P}\left({}^{\circ}C\right);\;R^{2}=0.93$$
(10)$$\sigma_{S}/\sigma_{S0}\left(\%\right)=-74+0.32\times T_{P}\left({}^{\circ}C\right);\;R^{2}=0.96$$
(11)$$\varepsilon_{R}/\varepsilon_{R0}\left(\%\right)=-69-0.07\times T_{P}\left({}^{\circ}C\right);\;R^{2}=1.00$$

Figure 10 compares the mechanical results of the present study with data reported from the literature and from the own authors experiments on different feedstock materials [8,25,27,28,29,30,31,32,33,34,35,36,37,38,39,40,41]. The performance of PLA is found among the top ranked materials. For instance, the stiffness of tested PLA is found within the range of PLA-based composites. The strength of 3D-printed PLA is also comparable to 3D-printed PLA-PHA and PLA-Hemp. A larger number of materials exhibit similar fracture toughness including PETG, PLA-Wood, PLA-PHA and PLA-Hemp. In terms of elongation at break, nylon is by far the best ranked material. It is followed by PETG and copolyester. PLA lies within the middle range. 

### 4.3. Microstructural Features

Fracture patterns of 3D-printed PLA are discussed through SEM micrographs in Figure 11. Alteration of filament compactness is limited and the fracture can be qualitatively related to filament tearing and stretching. 

The micrographs reveal the presence of microscopic aligned porosities of typical dimensions between 60 and 90 µm (the direction of alignment is highlighted in Figure 11). The stretching of the solid phase around these process-induced porosities can be as large as 440 µm. The observation of the filament geometry particularly within the external frame reveals consistency with the imposed layer height (0.2 mm). The filament section exhibits typical minor and major axis lengths of 0.13 mm and 0.21 mm, which represent a transverse shape factor of 0.62. The limited alteration of the filament section is a sign of more brittle fracture. Although the rough fracture surfaces indicate some plastic deformation, it appears that brittle fracture is the leading mechanism for fracture. 

The internal microstructural arrangement of 3D-printed PLA is discussed at the light of the X-ray micro-tomography results shown in Figure 12 for a sample printed at 230 °C. Figure 12a shows cross-section views normal to the three main directions. Two types of porosities can be associated to the processing. 

The first one forms underneath the external frame. This type of porosity originates from the abrupt change of the nozzle trajectory. This variation in trajectory and speed of material laying down creates the first type of material discontinuity. The second one is formed within the raster from the +45°/−45° sequencing of PLA filaments. The longitudinal view (plane XY) demonstrates a limited porosity amount organised in a grid-like structure with particular orientations at +45°/−45°. The porous structure induced by processing has a limited connectivity within this plane compared to the other orthogonal planes YZ and XZ. In particular, the porosity seems to be fully connected in Z-direction, which represents the building direction. The pore connectivity is entirely captured from perspective view by isolating the feature of interest in Figure 12b. From the same perspective view, the roughness generated by the printing of the external frame is also a distinctive feature. 

Figure 13 shows more quantitative information about the porosity arrangement through the axial porosity profiles. The overall porosity represents a content of 5.73% for a printing temperature of 230 °C. The same quantity derived from weight and volume measurements shows that this value is close to 11%.

There is, in general, a large difference between density-based and X-ray microtomography-based measurements of the porosity, although the two methods reveal the same tendencies with the increase of printing temperature. 

For density-based measurement, the following linear correlation is achieved
(12)$$f_{D}\left(\%\right)=24.5-0.06\times T_{P}\left({}^{\circ}C\right);\;R^{2}=0.92$$

For microtomography-based measurement, the linear approximation of the porosity content- printing temperature gives
(13)$$f_{T}\left(\%\right)=19.2-0.06\times T_{P}\left({}^{\circ}C\right);\;R^{2}=0.96$$

Such discrepancy can be attributed to two main reasons: -a slight overlap between the grey levels due to the solid phase and voids, especially around the void contours, influences the result of image processing;-the density measurements are performed on the entire sample whereas the image processing is conducted on the central part of the sample.

The highly jagged profiles along the sample width and length indicate a periodicity of the porosity positions and is consistent with a grid-like porous structure. Along the depth, the presence of large peaks at the first and last 20% of depth positions is informative about the amount of surface porosity that forms as a consequence of the filament crossing at the top and bottom layers as shown from the perspective views in Figure 12b. The packing of the PLA filaments is lower than the core, especially at the top surface (Figure 12) because the filaments are not forced to half of their height. Within the bulk of the specimen, the porosity level is lower and stable because of the layering effect. 

### 4.4. Predicted Tensile Behaviour

The effect of sampling on the predicted stress component $\sigma_{xx}$ distribution is shown in Figure 14. For a sample size of 0.4 mm equivalent to the nozzle dimension, the stress distribution varies significantly depending if a porosity feature is present within the volume. If so, the stress concentration is captured around the porosity as shown in Figure 14. 

For sample dimensions larger than 1.4 mm, the effect of porosity alignment tends to connect regions of higher stresses. Up to 2.9 mm, there is evidence of alternation between high and low stress fields that fully connect the entire sample. This large stress heterogeneity is due to the significant connectivity within the porosity network. The predicted Young’s modulus $E_{P}$ is linearly correlated to the sample size $D_{P}$ as long as this one is small then 3 mm. This correlation can be written as follows:(14)$$E_{P}\left(MPa\right)=1099-13.76\times D_{P}\left(mm\right);\;R^{2}=0.97$$

The predicted Young’s modulus becomes scale independent from a characteristic dimension of 7 mm. At the full scale, the effect of the resolution on the predicted stress component $\sigma_{xx}$ distribution is depicted in Figure 15.

The effect of the raster on the stress heterogeneity is captured for voxel sizes as large as 143 µm. This effect can be quantified as an alternation of low and stress levels along the inter-filament spaces. In the YZ plane, the stress heterogeneity marked by the layering effect is only observed for voxel sizes smaller than 72 µm. The predicted Young’s modulus is found to be nonlinearly and weakly dependent on the voxel size $d_{P}$.
(15)$$E_{P}\left(MPa\right)=977-25\times exp\left(-d_{P}\left(mm\right)/65\right);\;R^{2}=0.96$$

For a printing temperature of 230 °C, the predicted modulus at a voxel size of 24 µm is 959 MPa. The difference between the predicted and experimental Young’s modulus represents 21%. The predicted model overestimates the stiffness of the 3D-printed PLA under the hypothesis of a linear elasticity model. This overestimation can be related to the quality of the bond between the PLA filaments and represents about 25% with respect to the experimental stiffness of 3D-printed PLA. In addition, there is no obvious correlation between this discrepancy and the printing temperature. A proper evaluation of the mechanical properties of the interlayer bonding can be attempted to derive the influence of lack of load transfer between filaments on the overall performance. This can be performed using either particular mechanical setup such as pull up tests [42] or local mechanical tests such as nanoindentation [43]. The method considered in this study provides an estimated correction of the material stiffness based on finite element computation. The correction of intrinsic Young’s modulus $E_{Y0}$ is considered based on an inverse approach, which takes into account the contribution of the interfacial stiffness. The inverse approach relies on matching the slope of the predicted material response with the experimental one by adjusting $E_{Y0}$. This value is implemented in the FEM model to assess the full response of 3D-printed PLA, which is discussed hereafter. 

Figure 16 shows the predicted tensile response of 3D-printed PLA as a function of the printing temperature under the hypothesis of the linear damage model. Figure 16a depicts typical responses for 3D-printed PLA at 210 °C for various combinations of damage parameters. The damage model is capable of capturing the nonlinearity leading to material failure. 

Under the hypothesis of the linear damage model, this nonlinearity neglects plasticity effects and attributes the failure to damage accumulation. In order to compensate for the lack of inter-filament cohesion, the stiffness of PLA ($E_{Y0}$) is adjusted to fit the experimental slope in Figure 16a. The optimal value of 910 MPa is obtained (Table 2), which represents a 17% decrease from the experimental value obtained from filament tensile testing. 

Depending on the critical engineering strain level $\varepsilon_{r}$, damage onset can be delayed or prompted. From Figure 16a, the optimal value lies between 6.3% and 6.8% for a printing temperature of 210 °C. Both the modulus reduction ratio at the damaged zones ($E_{D}$) and damage rate $D_{r}$ control the tensile strength and the elongation at break. The nonlinear decrease of the tensile stress is represented by the optimal combination (0.5%, 2%) for the printing temperature of 230 °C. 

Figure 16b shows the evolution of the stress component σ_xx_ distribution for an increasing engineering strain under linear damage model hypothesis. The following parameters are used for this simulation: $T_{P\;}$= 210 °C, $E_{Y0}$ = 910 MPa, $\varepsilon_{r}$ = 6.5%, $E_{D}$ = 0.5%, $D_{r}$ = 2%. 

The simulation shows the localisation occurring at the central part of the specimen leading to a significant decrease in the stress level at large strain levels (typically $\varepsilon >\varepsilon_{r}$). The stress localisation occurs in a reduced area from the entire gauge length, which is in line with the experimental observations by the high-speed camera (Figure 7). 

Figure 16c shows the predicted tensile response of 3D-printed PLA for three printing temperatures. Each prediction is the result of an identification procedure requiring several finite element runs. The printed sample at 210 °C shows a larger capability for damage accumulation as attested by the damaged volume ratio that reaches in this case up to 5%. The reduction in the damage accumulation is concomitant with the improvement of the printing temperature. Samples printed above 210 °C show low damage accumulation of the order of 2%. The reduction in the damage volume is in agreement with the observed mechanical instability which leads to fast cracking behaviour instead of a more diffuse damage process. 

## 5. Conclusions 

The laying down process of PLA filament in a typical FDM process occurs with rapid thermal cycling and cooling rates as large as 8 °C/s. Within the range of printing temperature (210–230 °C), the linear improvement of tensile properties such as stiffness and strength and toughness are balanced by the linear decrease in the elongation at break when the printing temperature is increased. A significant loss of elongation at break of 3D-printed PLA compared to the as-received filament is the main feature attributed to the processing. A limited plasticity and instable cracking with crack speed of 440 m/s are other secondary features. Because of quasi-brittle failure, there is no obvious correlation between the raster configuration and the crack propagation path. The compactness of the filament arrangement with porosity as small as 12% contrasts with the large connectivity between porosities especially through the building direction. The predicted tensile performance can be considered as reliable for a characteristic length of 7 mm. The quality of the prediction becomes compromised if microstructural details as small as 72 µm are not accounted for in the finite element model. The linear elasticity model overestimates the stiffness of 3D-printed PLA by 25%, irrespective of the printing temperature, which questions the quality of the perfect inter-filament bonding assumed in the computation. The linear damage model captures the stress reduction in 3D-printed PLA and attributes the failure of the material to small and localised damage accumulation rather than to diffuse damage process. 

## Figures and Tables

**Figure 1 polymers-12-01060-f001:**
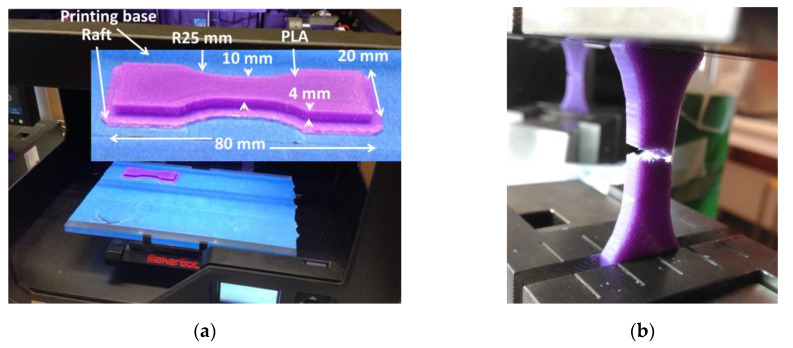
Tensile specimen printed using FDM: (**a**) Sample dimensions; (**b**) result of tensile testing.

**Figure 2 polymers-12-01060-f002:**
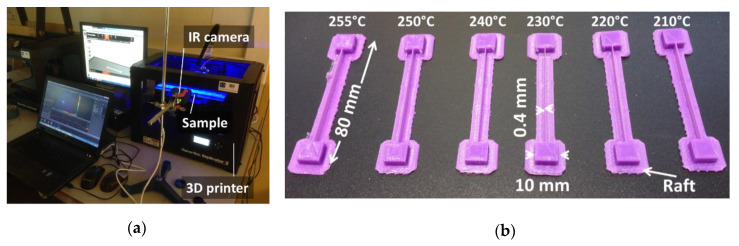
Infra-red measurements: (**a**) Setup; (**b**) produced samples for all printing temperatures.

**Figure 3 polymers-12-01060-f003:**
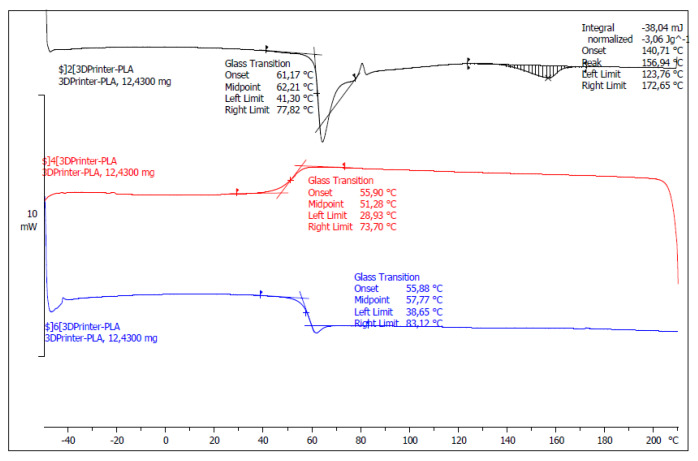
Thermal transitions of PLA filament assessed using two heating and one cooling run in DSC.

**Figure 4 polymers-12-01060-f004:**
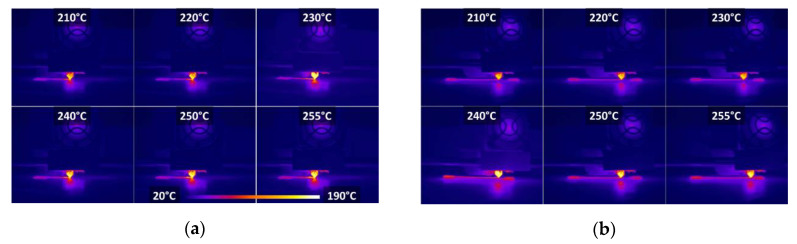
Infra-red recording of PLA temperature: (**a**) after the first 14 s and at; (**b**) the end of the printing sequence as a function of the printing temperature.

**Figure 5 polymers-12-01060-f005:**
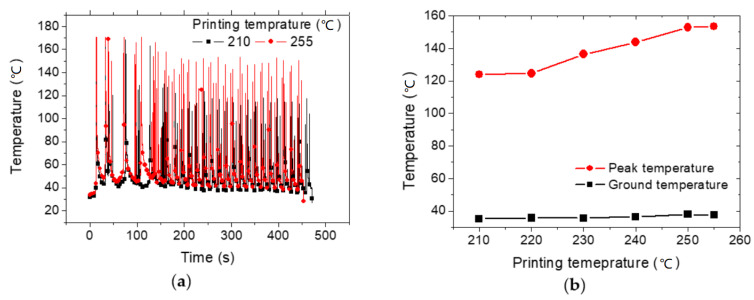
Quantitative infra-red results: (**a**) Thermal cycling for two different printing temperatures; (**b**) Average ground and peak temperatures for all printing conditions.

**Figure 6 polymers-12-01060-f006:**
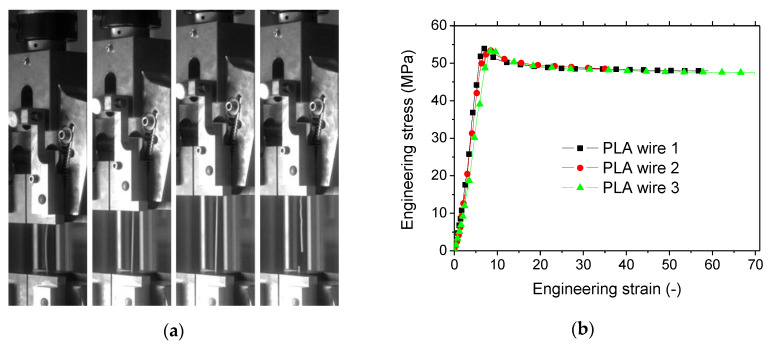
Tensile properties of as-received PLA wire: (**a**) Optical recording of deformed PLA wire under tension; (**b**) Tensile response of three replicates.

**Figure 7 polymers-12-01060-f007:**
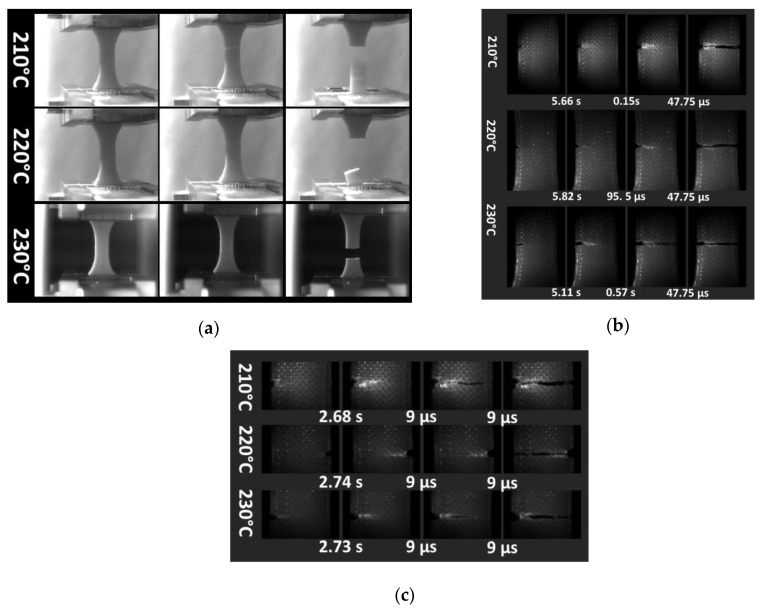
High speed camera recording of tensile test for: (**a**) neat and; notched 3D-printed PLA specimens at (**b**) intermediate, and (**c**) high speed observations.

**Figure 8 polymers-12-01060-f008:**
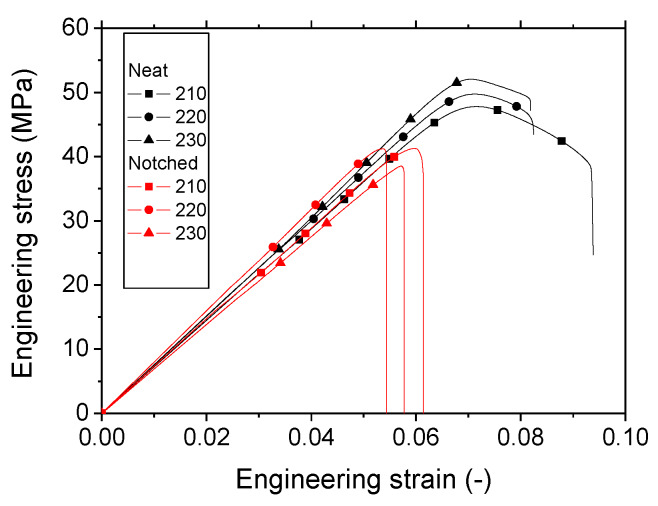
Tensile response of neat and notched 3D-printed PLA as a function of the printing temperature.

**Figure 9 polymers-12-01060-f009:**
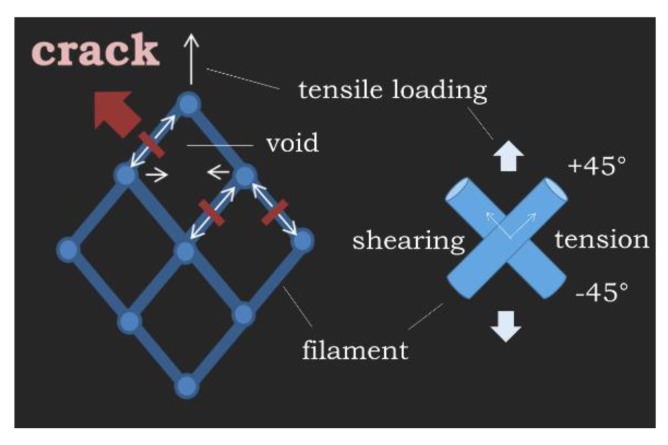
Stress transfer in 3D-printed PLA highlighting qualitative explanation of the difference between tensile properties of as-received and printed PLA.

**Figure 10 polymers-12-01060-f010:**
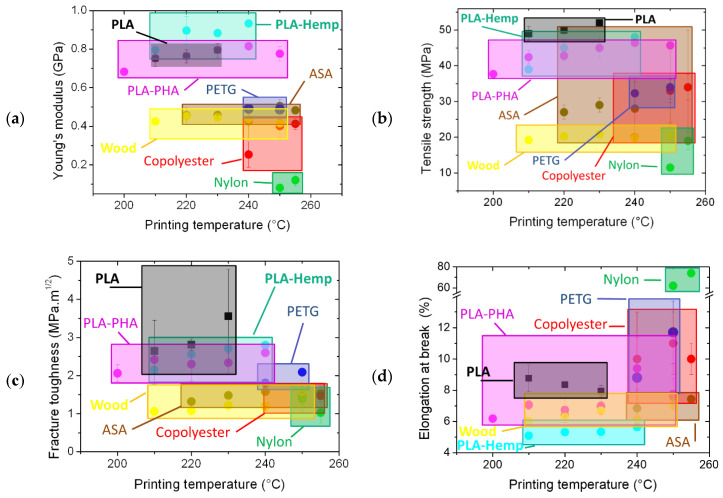
Tensile properties of eight different feedstock materials for FDM including PLA used in this study: (**a**) Young’s modulus, (**b**) tensile strength, (**c**) fracture toughness, and (**d**) elongation at break.

**Figure 11 polymers-12-01060-f011:**
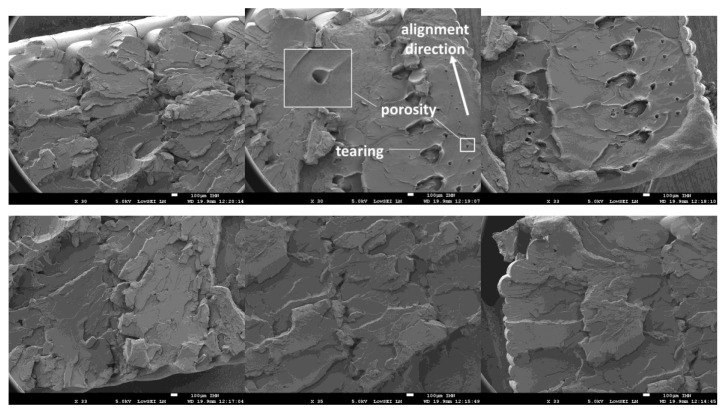
SEM images of fractured 3D-printed PLA (**top left**: fractured filament pattern, **top middle**: tearing and porosity alignment, **top right**: zoom out view of tearing pattern, **bottom left**: rough fractured surface, **bottom middle**: zoom in view of rough fractured surface, **bottom right**: fracture pattern at edges).

**Figure 12 polymers-12-01060-f012:**
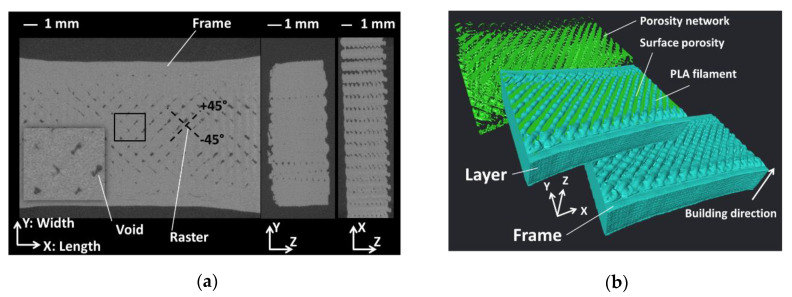
Main microstructural features of the 3D-printed PLA revealed using X-ray micro-tomography: (**a**) cross-section views; (**b**) porosity network and filament arrangement.

**Figure 13 polymers-12-01060-f013:**
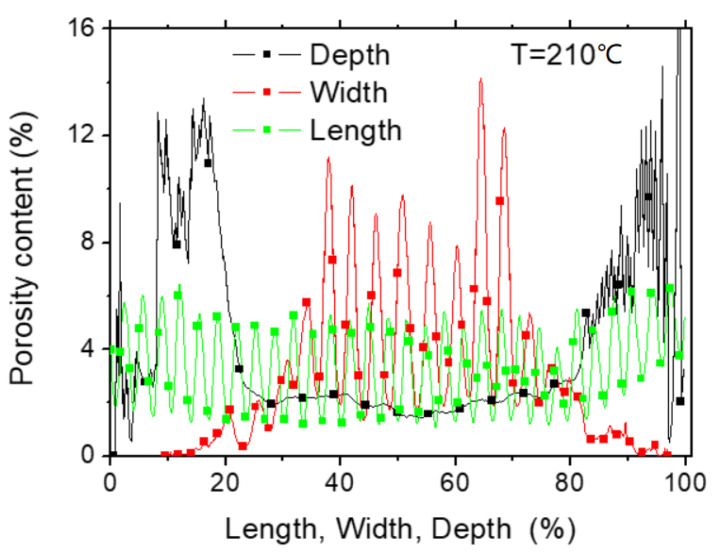
Evolution of the porosity content in the main orthogonal directions of 3D-printed PLA (sample dimensions 17 × 13 × 4 mm^3^).

**Figure 14 polymers-12-01060-f014:**
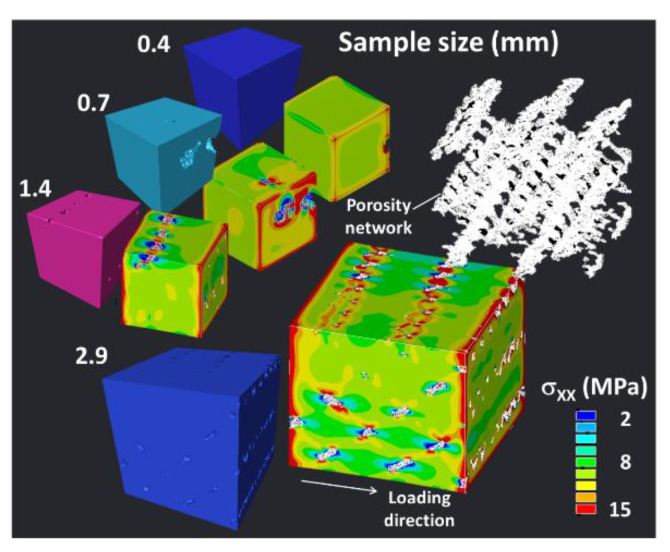
Effect of sample size on the stress component σ_XX_ distribution (voxel size is fixed to 14.31 µm, X is the loading direction).

**Figure 15 polymers-12-01060-f015:**
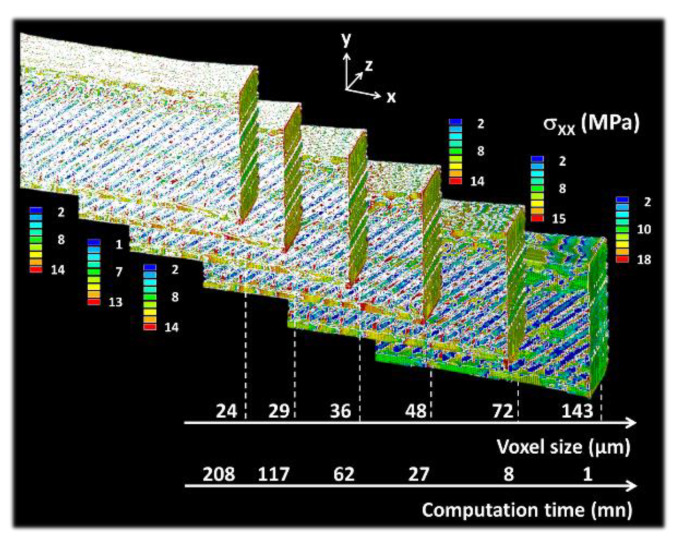
Effect of increasing the voxel size on stress component σ_XX_ counterplots (large voxel size represents a low resolution, X is the loading direction).

**Figure 16 polymers-12-01060-f016:**
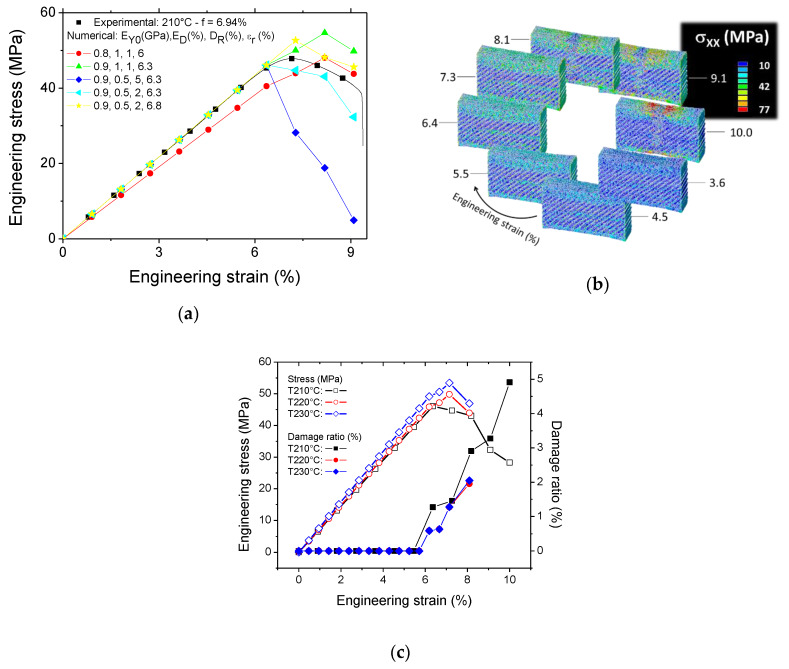
Damage modelling results: (**a**) Predicted tensile responses of 3D-printed PLA for various combinations of damage model parameters and a printing temperature of 210 °C, (**b**) typical predicted stress component σ_xx_ counterplot for an increasing engineering strain, and (**c**) tensile response as a function of printing temperature using linear damage model.

**Table 1 polymers-12-01060-t001:** Main mechanical parameters of as received wire and 3D-printed PLA as a function of printing temperature.

Material	Printing Temperature /°C	Density (g/cm^3^)	Porosity Content ^1^ (%)	Young’s Modulus (MPa)	Tensile Strength (MPa)	Elongation at Break (%)	Fracture Toughness MPa×m^1/2^
wire	-	1.24	0	1092 ± 136	54 ± 0 ^2^	55 ± 18	-
printed	210	1.09 ± 0.01	12.0 ± 1.1	751 ± 41	49 ± 2	8.77 ± 0.86	2.65 ± 0.81
	220	1.10 ± 0.02	11.1± 1.9	764 ± 32	50 ± 1	8.36 ± 0.16	2.81 ± 0.15
	230	1.11 ± 0.02	10.8± 1.92	794 ± 36	52 ± 1	7.95 ± 0.34	3.56 ± 1.23

^1^ Based on density measurements, ^2^ 0.14.

**Table 2 polymers-12-01060-t002:** Predicted mechanical parameters of 3D-printed PLA as a function of printing temperature.

Material	Printing Temperature /°C	Porosity Content ^1^ (%)	$E_{Y0}{}^{2}$ (MPa)	$\varepsilon_{r}$(%)	$E_{D}$ (%)	$D_{r}$ (%)
printed	210	6.94	910	49 ± 2	8.77 ± 0.86	2.65±0.81
	220	6.13	910	50 ± 1	8.36 ± 0.16	2.81±0.15
	230	5.77	910	52 ± 1	7.95 ± 0.34	3.56±1.23

^1^ based on X-ray micro-tomography, ^2^ adjusted.

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
