# Peer review of "On the Tensile Behaviour of Bio-Sourced 3D-Printed Structures from a Microstructural Perspective"

_polymers, 2020, doi:10.3390/polym12051060_

Round 1

Reviewer 1 Report

This research discussed the fabrication of PLA via 3D printing techniques and the analysis of their mechanical behaviors through the micromechanical analysis. FEM was also used to validate the mechanical behavior-microstructural feature relationships. 

  1. what are the main reasons that cause the crystallization behavior of PLA printed parts and their amorphous structures after heating?
  2. one important parameter that the author should have studied will be the interfacial interactions during printing - that means the dimensions of the printed parts should be managed and increased little by little. The printed lines, the printed areas with a single layer and the printed parts of different layers should be studied - are they consistent? The void distribution and the porosity should influence their mechanical properties.
    The temperature influence on the mechanical properties will be incremental. does thermal treatment to get rid of the voids help in their mechanical performance?

Author Response

Reviewer #1.

This research discussed the fabrication of PLA via 3D printing techniques and the analysis of their mechanical behaviors through the micromechanical analysis. FEM was also used to validate the mechanical behavior-microstructural feature relationships. 

We thank the reviewer for this clear introduction to our work. Here is below our answer to his comments.

  1. what are the main reasons that cause the crystallization behavior of PLA printed parts and their amorphous structures after heating?

This is a very interesting question that we did not properly addressed in this work because the main focus was on the microstructural influence on the performance of printed parts. We rely for this answer on the recent work published by Liao et al. 2019, which shows several key facts about the crystallisation behaviour. The first one is that the semi-crystaline PLA when melted exhibits different trends depending on the thermal conditions. It is, for instance, known that the printing temperature and the base temperature greatly influence the relaxation phenomena. Since the crystallisation process is a slow process compared to the extrusion, the authors found that some instable forms of crystalline (delta form) emerges for thick printed samples. The authors correlated the presence of delta form to the lowering of the stiffness of the printed parts. In addition, Pandis et al. showed that the increase of crystallinity is found after several days of in-service use for 3D printed PLA. The authors attributed this phase change to material aging.

Amendment in page 2: « It is well known that printing… stiffness of printed PLA ».

References added :

Liao, Y., Liu, C., Coppola, B., Barra, G., Di Maio, L., Incarnato, L., Lafdi, K., 2019. Effect of Porosity and Crystallinity on 3D Printed PLA Properties. Polymers 11, 1487.

Lee, C.-Y., Liu, C.-Y., 2019. The influence of forced-air cooling on a 3D printed PLA part manufactured by fused filament fabrication. Additive Manufacturing 25, 196-203.

Pandis, P.K., Papaioannou, S., Koukou, M.K., Vrachopoulos, M.G., Stathopoulos, V.N., 2019. Differential scanning calorimetry based evaluation of 3D printed PLA for phase change materials encapsulation or as container material of heat storage tanks. Energy Procedia 161, 429-437.

Guessasma, S., Belhabib, S., Nouri, H., 2019. Understanding the microstructural role of bio-sourced 3D printed structures on the tensile performance. Polymer Testing 77, 105924.

one important parameter that the author should have studied will be the interfacial interactions during printing - that means the dimensions of the printed parts should be managed and increased little by little. The printed lines, the printed areas with a single layer and the printed parts of different layers should be studied - are they consistent? The void distribution and the porosity should influence their mechanical properties.

We agree with the reviewer that the interfacial interaction is very important in tailoring the performance of printed parts. We already addressed this particular aspect in former contributions. In a former paper (Guessasma et al. Europ Pol J. 2016), we showed that scaling up the same print (from 5 mm up to 40 mm) led to varied mechanical properties. In particular, it was found that both stiffness and strength were lower for large samples compared to small ones because the former exhibited rapid cooling rates. We attributed this difference to a lower cohesion between adjacent filaments for large samples because of the rapid thermal cycling. 

Unfortunately, when we planned our experimental campaign we did not think of varying the layer thickness because this would result in a gradient of properties that we wanted to avoid. However, we studied, in a former work (Guessasma et al. Polymer 2015) through X-ray micro-tomography, the void distribution along the main directions and we observed different porosity arrangements depending on the orientation of the filaments. These different arrangements of the porosity network led to varied stiffness properties including Young’s modulus and Poisson’s ratio. As a complementary remark, Chacon et al. studied the effect of layer thickness on the mechanical performance of 3D printed PLA. The authors concluded on a contrasted effect of this parameter depending on the part orientation. The most significant effect was related to the ductility decrease when the layer thickness was increased.

Amendment: page 2: “One of the particular aspect that received raster and the thermal history.

Reference added:

Chacón, J.M., Caminero, M.A., García-Plaza, E., Núñez, P.J., 2017. Additive manufacturing of PLA structures using fused deposition modelling: Effect of process parameters on mechanical properties and their optimal selection. Materials & Design 124, 143-157.

Guessasma, S., Belhabib, S., Nouri, H., 2015. Significance of pore percolation to drive anisotropic effects of 3D printed polymers revealed with X-ray μ-tomography and finite element computation. Polymer 81, 29-36.

Guessasma, S., Belhabib, S., Nouri, H., Ben Hassana, O., 2016. Anisotropic damage inferred to 3D printed polymers using fused deposition modelling and subject to severe compression. European Polymer Journal 85, 324-340.

The temperature influence on the mechanical properties will be incremental. does thermal treatment to get rid of the voids help in their mechanical performance?

Yes, several studies demonstrate that post treatment of 3D printed parts improves the mechanical performance of 3D printed parts. For instance, Shaffer et al. reported that improved interlayer adhesion is obtained when the 3D printed part is exposed to ionizing radiation. This treatment contributed to enhance the polymer crosslinking. Also, Bhandari et al. showed that annealing treatment with temperatures higher than the glass transition temperature improves the inter-layer tensile strength.

Added reference

Bhandari, S., Lopez-Anido, R.A., Gardner, D.J., 2019. Enhancing the interlayer tensile strength of 3D printed short carbon fiber reinforced PETG and PLA composites via annealing. Additive Manufacturing 30, 100922.

Shaffer, S., Yang, K., Vargas, J., Di Prima, M.A., Voit, W., 2014. On reducing anisotropy in 3D printed polymers via ionizing radiation. Polymer 55, 5969-5979.

Change in page 2: “Because of the importance of the ... the inter-layer tensile strength.”.

Reviewer 2 Report

In this manuscript, the authors analysed the influence of the microstructural arrangement of PLA filaments on tensile behaviour of printed parts by using experimental and numerical approaches. The work is well written even if the characterization of PLA printed part is something already seen in literature both from experimental and numerical point of view. Mechanical characterization are well conducted and the results sound interesting. However, the work lack of innovation even if the experimental and simulation part have been well conducted. The correlation between printing temperature and mechanical properties sound quite interesting even if the results did not highlight a high variation in mechanical properties with the printing temperature. I would suggest the acceptance of the present manuscript in Polymers journal after some major revision that I reported below:

  • At p. 2 line 56 there is a spelling error. Please substitute “to quantity..” with “to quantify..”.
  • I would suggest expanding the introduction part by citing other works in literature, that talk about 3D printed PLA samples characterization. I would also suggest highlighting the difference between the author’s approach and other approach present in literature.
  • The authors used DSC analysis to assess the glass transition (Tg) and melting (Tm) temperatures of the PLA wire. However, it is not clear why did they choose to use the second scan to perform the measure. In fact, in the second scan, the thermal history of the wire is lost and so they are characterizing another kind of material not the raw PLA filament that they will use in the 3D printing process.
  • The authors varied the printing temperatures in range from 210 to 255 °C. However, in my opinion 255 is too high as processing temperature and maybe the PLA could start degradation process at that temperature. I would suggest to the authors to perform a thermogravimetric analysis (TGA) in air to understand at which temperature the PLA start the degradation process.
  • At p. 2 line 79 the authors asses that the main dimension of dog-bone specimen are shown in figure 1b. However, form figure 1b it is impossible to have quantitative information about the specimen dimensions.
  • At p.3 line 114 I the meaning of the sentence “the solid phase is regularly meshed, accordingly” is not clear. “Accordingly“ to what?
  • At p.3 line 120 please correct the apex format of the unit of the sample volume.
  • At p.4 line 149 the authors assessed that “Kim et al. [10] reported slightly larger temperatures for Tg (62.9°C) and melting (167.2°C) for PLA similar to the one considered in this study”. I think that it is the opposite. The value of Kim et al are similar for Tg and slightly higher for melting.
  • Figure 4 is difficult to understand. I mean that the sense is clear but the infra-red camera has low resolution and so it is not possible to understand the differences in material temperature suddenly after deposition for the different printing temperature considered.
  • It is not clear what does Figure 5a represent. Please explain better what the authors want to show in the Figure.
  • The authors reported different mechanical behaviour between PLA wires and PLA 3D printed samples. Do you have an explanation for so different behaviour?
  • In Figure 9, the authors reported tensile properties of eight different feedstock materials for FDM including PLA. However, among tensile properties they did not report the elongation at break (%) as a function of printing temperatures. Please report the same kind of plot also for the elongation at break.
  • The authors assessed that figure 10 reveal the presence of microscopic aligned porosities of typical dimensions 253 between 60 μm and 90 μm. However, it is difficult from these images to assess the presence of an open porosity typical of 3D structures printed buy FDM. Please select different images in which is clear the presence of an open porosity inside the structure.
  • At p. 12 line 335 the authors asses that the predicted model overestimates the stiffness of 3D printed PLA samples and this overestimation can be correlated to the quality of the bond between PLA layers. Did you evaluate the mechanical properties of the interlayer bonding? You can perform some mechanical test to evaluate the mechanical properties of the bonding between layer and us this value in the FEM model.

Author Response

Reviewer #2.

In this manuscript, the authors analysed the influence of the microstructural arrangement of PLA filaments on tensile behaviour of printed parts by using experimental and numerical approaches. The work is well written even if the characterization of PLA printed part is something already seen in literature both from experimental and numerical point of view. Mechanical characterization are well conducted and the results sound interesting. However, the work lack of innovation even if the experimental and simulation part have been well conducted. The correlation between printing temperature and mechanical properties sound quite interesting even if the results did not highlight a high variation in mechanical properties with the printing temperature. I would suggest the acceptance of the present manuscript in Polymers journal after some major revision that I reported below:

We thank the reviewer for his positive opinion and his interesting remarks. He will find hereafter our detailed answer.

  • At p. 2 line 56 there is a spelling error. Please substitute “to quantity..” with “to quantify..”.

We are sorry for this lingual error. This was corrected in the new version.

  • I would suggest expanding the introduction part by citing other works in literature, that talk about 3D printed PLA samples characterization. I would also suggest highlighting the difference between the author’s approach and other approach present in literature.

We already expanded the literature work based on the remarks of the first reviewer. We also managed to add few words to highlight the relevance of our approach compared to the literature work.

Amendment in pages 2-3 “The approach considered”… not clearly obvious. »

Added references:

Sood, A.K., Ohdar, R.K., Mahapatra, S.S., 2009. Improving dimensional accuracy of Fused Deposition Modelling processed part using grey Taguchi method. Materials & Design 30, 4243-4252.

Yu, S., Hwang, Y.H., Hwang, J.Y., Hong, S.H., 2019. Analytical study on the 3D-printed structure and mechanical properties of basalt fiber-reinforced PLA composites using X-ray microscopy. Composites Science and Technology 175, 18-27.

  • The authors used DSC analysis to assess the glass transition (Tg) and melting (Tm) temperatures of the PLA wire. However, it is not clear why did they choose to use the second scan to perform the measure. In fact, in the second scan, the thermal history of the wire is lost and so they are characterizing another kind of material not the raw PLA filament that they will use in the 3D printing process.

The reviewer is right. We use the first scan to characterise the thermal properties of the raw material. See for instance the discussion of melting and glass transition temperatures from the first run depicted in page 5 “Figure 3 shows that DSC measurements…”. But also, we use the second scan to mimic thermal changes after erasing the thermal history of the filament. This is a useful information after printing (even if the thermal changes are significantly amplified by the rapid laying down process). We clarified this aspect in the new version.

Amendment in page 3: “The first scan is used to …after erasing the thermal history of the filament.

  • The authors varied the printing temperatures in range from 210 to 255 °C. However, in my opinion 255 is too high as processing temperature and maybe the PLA could start degradation process at that temperature. I would suggest to the authors to perform a thermogravimetric analysis (TGA) in air to understand at which temperature the PLA start the degradation process.

As stated in our former version, the purpose of using high temperature is to cover the mechanical performance for printing temperatures that were not reported in the literature and establish a precise parameter window for PLA printing.

See for instance in page 3: “Although, the recommended printing temperatures for PLA are below 230°C, the objective of using larger temperatures is to assess the effect of large temperature ranges on mechanical performance that are not yet reported.”

Amendment in page 3: “This allows to establish a more precise parameter window for PLA printing.”  

We did not conduct TGA analysis but we rely on the results from Coppola et al. which shows that degradation of PLA starts at higher temperatures.

Amendment in page 6: “Although, the low limit for printing temperature used in this study (210°C) is commonly used for PLA, high processing temperatures such as 255°C may result in PLA degradation process. Coppola et al.(Coppola et al., 2018) conducted a thermogravimetric analysis (TGA) up to 700°C, which shows weight loss values of 25% at temperatures starting from 346°C. Their results suggest that the thermal stability of PLA is not compromised at 255°C. Silverajah et al. (Silverajah et al., 2012), in an earlier work, report a slightly lower degradation temperature for PLA of 325°C.”

References added

Silverajah, V.S.G., Ibrahim, N.A., Yunus, W.M.Z.W., Hassan, H.A., Woei, C.B., 2012. A Comparative Study on the Mechanical, Thermal and Morphological Characterization of Poly(lactic acid)/Epoxidized Palm Oil Blend. International Journal of Molecular Sciences 13, 5878-5898.

  • At p. 2 line 79 the authors asses that the main dimension of dog-bone specimen are shown in figure 1b. However, form figure 1b it is impossible to have quantitative information about the specimen dimensions.

The reviewer is right. The main dimensions of dog bone specimens are shown in Figure 1a not in figure 1b. Correction is made accordingly.

  • At p.3 line 114 I the meaning of the sentence “the solid phase is regularly meshed, accordingly” is not clear. “Accordingly“ to what?

The sentence was rephrased to highlight the fact that regular meshing is related to the voxel-based images.

  • At p.3 line 120 please correct the apex format of the unit of the sample volume.

We are sorry for this wrong formatting. It is now corrected. Also the magnitude of the number of degrees of freedom is now corrected.

  • At p.4 line 149 the authors assessed that “Kim et al. [10] reported slightly larger temperatures for Tg (62.9°C) and melting (167.2°C) for PLA similar to the one considered in this study”. I think that it is the opposite. The value of Kim et al are similar for Tg and slightly higher for melting.

Considering the percentage of variation between our data and the ones reported by Kim et. al (1% for Tg and 6% for Tm) the reviewer is right. The sentence was corrected.

  • Figure 4 is difficult to understand. I mean that the sense is clear but the infra-red camera has low resolution and so it is not possible to understand the differences in material temperature suddenly after deposition for the different printing temperature considered.

We agree with the reviewer. This is true because the resolution of the camera is only 320 x 256 pixels. However, we report only that the rapid cooling can be clearly seen from the contrast between the nozzle temperature (290°C) and the deposited laid down filament (100°C).

Amendment in pages 5 – 6 :” Despite the limited resolution of… temperature reading are used. ”

  • “It is not clear what does Figure 5a represent. Please explain better what the authors want to show in the Figure.

We agree with the reviewer that the explanation of the thermal cycles was limited in the former version. In the new one, we added more explanation.

Amendment in page 6: “To build the thermal cycle profiles in Figure 5a, … the filament laying down.    ”

  • The authors reported different mechanical behaviour between PLA wires and PLA 3D printed samples. Do you have an explanation for so different behaviour?

The filament placing in a sequence of +45/-45 allows only a limited part of the filament to stretch. At these locations the PLA filament achieves its elongation at break. This effect is more amplified for filaments with larger ultimate properties such as copolyester. A new figure is added to qualitatively explain this correlation. A more elaborated discussion of the difference between the as-received material behaviour and the printed part perfprmance is also added in the text.

Amendment in page 9: “The comparison between the as-received and printed PLA performance shows that …to its elongation at break. This effect is more amplified for filament with larger ultimate properties such as copolyester (Abouzaid et al., 2018).   ”

Added reference:

Abouzaid, K., Guessasma, S., Belhabib, S., Bassir, D., Chouaf, A., 2018. Printability of co-polyester using fused deposition modelling and related mechanical performance. European Polymer Journal 108, 262-273.

  • In Figure 9, the authors reported tensile properties of eight different feedstock materials for FDM including PLA. However, among tensile properties they did not report the elongation at break (%) as a function of printing temperatures. Please report the same kind of plot also for the elongation at break.

We thank the reviewer for this remark. Indeed, as the most significant effect is related to the decrease of the elongation at break, we worked on this particular aspect to highlight the difference between feedstock materials. As requested by the reviewer we also reported the graph for the elongation at break.

Major change: Figure 9d added and renumbering of all following figures + text modification in page 10 “In terms of elongation at break, … within the middle range. ”

  • The authors assessed that figure 10 reveal the presence of microscopic aligned porosities of typical dimensions 253 between 60 μm and 90 μm. However, it is difficult from these images to assess the presence of an open porosity typical of 3D structures printed buy FDM. Please select different images in which is clear the presence of an open porosity inside the structure.

We highlighted the alignment of the porosities in SEM images by some labelling as we do not have additional SEM micrographs. In addition, Figure 12b shows such feature from X-ray micro-tomography more clearly than from SEM images.

Amendment: figure 10 (now Figure 11) modified + text added in page 11 “(the direction of …in figure 11).”

  • At p. 12 line 335 the authors asses that the predicted model overestimates the stiffness of 3D printed PLA samples and this overestimation can be correlated to the quality of the bond between PLA layers. Did you evaluate the mechanical properties of the interlayer bonding? You can perform some mechanical test to evaluate the mechanical properties of the bonding between layer and us this value in the FEM model.

We agree with the reviewer that complementary local tests can be used to evaluate the overestimation of stiffness. We could not perform these tests. However, we provided an estimated correction of the material stiffness based on finite element computation as shown in Figure 16a (former Figure 15a). In this figure the correction in Young’s modulus takes into account the interfacial effect.

Amendment in page 14: “A proper evaluation of the mechanical properties of the interlayer bonding can be attempted to derive the influence of lack of load transfer on the overall performance. This can be performed using either particular mechanical setup such as pull up tests (Frikha et al., 2017) or local mechanical tests such as nanoindentation (Guessasma et al., 2017). The method considered in this study provides an estimated correction of the material stiffness based on finite element computation. The correction of intrinsic Young’s modulus E_Y0 is considered based on an inverse approach which takes into account the contribution of the interfacial stiffness. The inverse approach relies on matching the slope of the material predicted response with the experimental one by adjusting E_Y0. This value is implemented in the FEM model to assess the full response of 3D printed PLA, which is discussed hereafter.”

Added references:

Guessasma, S., Zhang, W., Zhu, J., 2017. Local mechanical behavior mapping of a biopolymer blend using nanoindentation, finite element computation, and simplex optimization strategy. Journal of Applied Polymer Science 134.

Frikha, M., Nouri, H., Guessasma, S., Roger, F., Bradai, C., 2017. Interfacial behaviour from pull-out tests of steel and aluminium fibres in unsaturated polyester matrix. Journal of Materials Science 52, 13829-13840.

Round 2

Reviewer 1 Report

accept

Reviewer 2 Report

The authors replied to all the reviewer comments improving the scientific soundness of the manuscript. Actually some points have been clarified and some flaws have been corrected. Due to the substantial changes made by the authors I would suggest the acceptance of the manuscript in the present form.